# Impact of Europium and Niobium Doping on Hafnium Oxide (HfO$_2$): Comparative Analysis of Sol–Gel and Combustion Synthesis Methods

Katrina Laganovska *, Virginija Vitola, Ernests Einbergs, Ivita Bite, Aleksejs Zolotarjovs, Madara Leimane, Gatis Tunens and Krisjanis Smits

Institute of Solid State Physics, University of Latvia, Kengaraga Str. 8, LV-1063 Riga, Latvia; virginija.vitola@cfi.lu.lv (V.V.); ernests.einbergs@cfi.lu.lv (E.E.); ivita.bite@cfi.lu.lv (I.B.); aleksejs.zolotarjovs@cfi.lu.lv (A.Z.); madara.leimane@cfi.lu.lv (M.L.); gatis.tunens@cfi.lu.lv (G.T.); krisjanis.smits@cfi.lu.lv (K.S.)
* Correspondence: katrina.laganovska@cfi.lu.lv

**Abstract:** This study compares HfO$_2$ ceramics synthesized using sol–gel and combustion methods, emphasizing the impact of the method of synthesis on the resulting properties of the material. The research findings illustrate morphological differences between sol–gel and combustion-derived HfO$_2$. While sol–gel samples displayed irregular nanoparticles with pronounced boundaries, combustion samples revealed more homogeneous structures with particles tending towards coalescence. It was discerned that Eu$^{3+}$ doping induced oxygen vacancies, stabilizing the tetragonal phase, while subsequent doping with Nb$^{5+}$ significantly reduced these vacancies, which was also observed in photoluminescence analysis. Furthermore, combustion synthesis left fewer organic residues, with urea presence during synthesis contributing to residual organic components in the material. XPS analysis was used to evaluate the presence of oxygen-deficient hafnia sub-oxide in the samples. The study underscores the important role of tailored synthesis methods in optimizing the properties and applications of HfO$_2$.

**Keywords:** sol–gel; combustion; charge compensation; metal oxide; HfO$_2$; photoluminescence; XPS; oxygen vacancies

## 1. Introduction

Hafnium oxide (HfO$_2$ or hafnia) has emerged as a material of significant interest due to its versatile applications in microelectronics, optics, and catalysis, among other domains [1–3]. Hafnium oxide is commonly used in electronics because of its high dielectric constant. It has a wide band gap (>5 eV) [4], high refractive index [5], and has been used extensively as antireflection and optical coatings for interference filters [6,7]. HfO$_2$ is also a high-density material (~10 g/cm$^3$), and, when activated with rare earth elements (i.e., Eu$^{3+}$), acts as good scintillating materials and thermal barrier coatings for operation in harsh and high-temperature environments [8–12].

HfO$_2$ exists in different polymorph phases—cubic (space group: *Fm3m*), tetragonal (*P42/nmc*), orthorhombic (*Pca21*), and monoclinic (*P21/s*) [13]. The monoclinic phase is stable from RT up to 2000 K. The tetragonal phase is observed at 2000 K–2900 K. The highest symmetry cubic phase is stabilized between 2900 K and 3100 K. [13] The orthorhombic phase is typically not a stable phase under normal pressure conditions [14]. Various synthesis methods have been explored to optimize the physicochemical properties of HfO$_2$ and tailor its application-specific attributes [15–18]. Synthesizing hafnium oxide through various techniques allows for control over the particle size, morphology, and crystallinity of the hafnia, which in turn affects its properties and therefore is crucial for intentional tailoring of the material properties. When selecting a method of synthesis, properties such

as the desired particle size, morphology, and crystallinity, as well as defect control, cost, and ecological footprint need to be taken into consideration. Each method may offer specific advantages in terms of these parameters [19–21]. In this manuscript, we compare hafnia prepared by combustion and sol–gel methods. Both methods are popular for synthesizing advanced ceramics and have unique characteristics, and, with this publication, we want to emphasize that the method of synthesis plays a crucial role in the amount of oxygen vacancies, the structure of obtained ceramics, and, in turn, the usability of the ceramics. In addition to undoped $HfO_2$, the samples are also doped with $Eu^{3+}$ ions known to create oxygen vacancies [22–24] and then further doped with $Nb^{5+}$ ions which act as a charge compensating element that in turn reduces the amount of oxygen vacancies, allowing the exploration of intrinsic defects in a controlled manner [22,25,26]. $Eu^{3+}$ ions are also known as luminescent probes as their luminescence is directly related to the surrounding structure [25].

## 2. Materials and Methods

### 2.1. Materials

$HfCl_4$, purity 99.9%; $Eu_2O_3$, purity 99.99%; and $NbCl_5$, purity 99% were used as the starting materials for non-doped and doped $HfO_2$ and were purchased from Alfa Aesar, Germany. The synthesis description from Laganovska et al. [23] was used.

Citric acid anhydrous $C_6H_8O_7$, purity 99.0%, Lach:ner, Czech Republic; ethylene glycol ($HO(CH_2)_2OH$, purity 99.5%, Fisher Chemical, Belgium); glycine ($C_2H_5NO_2$, purity 99.7%, Sigma Aldrich, China); urea ($NH_2CONH_2$, purity 99.5%, Carl Roth, Germany); hexamethylenetetramine ($C_6H_{12}N_4$, HMTA, purity 99.0%, Chempur, Poland); and nitric acid ($HNO_3$, assay 70%, Sigma Aldrich, France) were used for the synthesis as described below. All the chemicals used in this work are reagent and analytical grade and used without further purification.

### 2.2. Synthesis

2.2.1. Sol–Gel Synthesis (SG)

1.1    SG-PC:

Deionized water was used to dissolve at room temperature (RT) under constant stirring speed for 10 min. An amount of 0.2 mol/L of $HfCl_4$ was added subsequently and stirring was continued for 10 min until $Hf_4$ was dissolved and ethylene glycol (EG) was added, maintaining a 1:4 citric acid and $HNO_3$ molar ratio.

1.2    SG-G:

Deionized water was used to dissolve $HfCl_4$ (0.2 mol/L) at RT with constant stirring speed for 10 min. After that, glycine was added (4 mol/L), maintaining a constant stirring rate for 10 more minutes. An appropriate amount of concentrated nitric acid ($HNO_3$) was added, maintaining a 1:43 glycine and EG molar ratio.

1.3    SG-U:

Deionized water was used to dissolve $HfCl_4$ 0.2 mol/L at RT with constant stirring speed for 10 min. An amount of 4 mol/L of $(NH_2)_2CO$ (urea) was added.

2    The synthesis in all cases was continued with the following:

The reaction was then carried out at 90 °C while maintaining the same constant stirring speed while a white solid gel had formed. The gel was placed in a vertical muffle furnace and heated at 400 °C for 2 h, the heating rate being 13 °C/min and allowing it to cool naturally. Calcination of the obtained sample in air was carried out at 800 °C for 2 h with a heating rate of 5 °C min and natural cooling.

For 5 atom % $Eu^{3+}$-doped $HfO_2$ and 5 atom % $Eu^{3+}$, 5 atom % $Nb^{5+}$-doped $HfO_2$ nanomaterial synthesis $Eu^{3+}$ and $Nb^{5+}$ ion solution was added dropwise to the $Hf^{4+}$ ion solution and stirred for 10 min before the addition of urea.

2.2.2. Combustion Synthesis (CO)

1    For all samples:

Deionized water was used to dissolve $HfCl_4$ (0.2 mol/L) at RT with constant stirring speed for 10 min.

1.1    CO-U:

A homogeneous Hf ion solution was obtained and 4 mol/L of urea was added under the same stirring conditions for another 10 min until a homogeneous metal–urea complex solution was obtained. Concentrated $HNO_3$ was added at room temperature, maintaining stirring, maintaining a 1:1.5 urea and $HNO_3$ molar ratio.

1.2    CO-H:

A homogeneous Hf ion solution was obtained and 4 mol/L of hexamethylenetetramine (HMTA) was added under the same stirring conditions, maintaining a 1:2 molar ratio of Hf ions and HMTA. Concentrated $HNO_3$ was added at room temperature, maintaining a 1:5 HMTA and $HNO_3$ molar ratio.

1.3    CO-UH:

A homogeneous Hf ion solution was obtained and 4 mol/L of hexamethylenetetramine (HMTA) was added under the same stirring conditions. The resulting mixture was stirred at room temperature for 10 min and then 4 mol/L of urea was added under the same stirring conditions for another 10 min. The molar ratios of Hf ions, HMTA, and urea were 1:2:2. Concentrated $HNO_3$ was added at room temperature, maintaining stirring. The molar ratios of HMTA:$HNO_3$ was 1:3 and urea:$HNO_3$ was 1:1.5.

2    The synthesis in all cases was continued with the following:

The reaction mixture was stirred at a constant speed for 10 min (pH = 3). Afterwards, the mixture was heated at 100 °C until white gel was formed. The gel was then placed in a preheated vertical muffle furnace and the auto combustion (SCS) reaction was carried out at 300 °C until rapid gas evolution and self-ignition process was observed. The obtained grayish-white powder was calcined at 800 °C for 2 h in air, with a heating rate of 5 °C min and natural cooling.

For 5 atom % $Eu^{3+}$-doped $HfO_2$, 5 atom % $Eu^{3+}$, and 5 atom % $Nb^{5+}$-doped $HfO_2$ nanomaterials, the appropriate amount of $Eu^{3+}$ and $Nb^{5+}$ ion solution was added dropwise to the $Hf^{4+}$ ion solution under constant stirring speed before the addition of urea/HMTA.

List of the methods of synthesis:

- SG—sol–gel synthesis
- SG-PC—sol–gel synthesis—polymerized complex
- SG-G—sol–gel synthesis—glycine
- SG-U—sol–gel synthesis—urea
- CO—combustion synthesis
- CO-U—combustion synthesis—urea
- CO-H—combustion synthesis—hexamine
- CO-UH—combustion synthesis—urea and hexamine

An amount of 250 MPa was then applied to the obtained nanoparticle powders and samples were pressed into 5 mm diameter green ceramic pellets with a thickness of 1 mm.

### 2.3. Characterization

The crystalline structure of the samples was determined using X-ray diffraction (XRD) with an X-ray diffractometer (X'Pert Pro MPD) with Cu-Kα radiation (λ = 1.54 nm). Crystalline sizes were determined from XRD data using the Scherrer equation. Luminescence measurements were performed using a YAG laser FQSS266 (CryLas GmbH) 4th harmonic at 266 nm (4.66 eV) at room temperature. All samples were pressed into tablets of equal size, which allowed for intensity comparison between the samples. The luminescence spectra were recorded using an Andor Shamrock B-303i spectrograph equipped with a CCD

camera (Andor DU-401A-BV). The FTIR spectra were measured with a Bruker—Equinox 55 Fourier transform infrared spectrometer with a frequency range of 370–25,000 cm$^{-1}$ and resolution >0.5 cm$^{-1}$. A scanning electron microscope (SEM), Thermo Fisher Scientific Helios 5 UX, was used to examine morphologies and microstructure of all samples in secondary electron (SE) mode. The elemental composition was studied using energy-dispersive X-ray spectroscopy (EDX). SEM was operated for imaging at 2 kV voltage and during the elemental analysis at 30 kV. To measure the XPS spectra, an XPS (ESCALAB 250 Xi) hemispherical analyzer was operated at 20 eV pass and using Al-K radiation, with an overall resolution of approximately 0.45 eV.

## 3. Results and Discussion

### 3.1. XRD and EDX

According to previous research, the phase of HfO$_2$ does not impact the luminescence intensity as much as the oxygen vacancy presence and distribution [25,26]. All samples were therefore synthesized with the aim to maintain the monoclinic phase. Figure 1 shows the X-ray diffraction patterns for the (a) undoped HfO$_2$, (b) HfO$_2$:Eu, and (c) HfO$_2$:Eu for Nb samples. The Eu-doped samples show a typical addition of a tetragonal phase, ranging from 6.0% to 9.1% (Table 1). When the samples are doped with Eu$^{3+}$, oxygen vacancies are created as a charge-compensating mechanism, leading to the stabilization of the tetragonal phase. Further, when the samples are additionally doped with Nb$^{5+}$, the Nb ions act as a charge-compensating element and the amount of oxygen vacancies is drastically reduced, resulting in a tetragonal phase percentage of a maximum of 4.4% (Table 1).

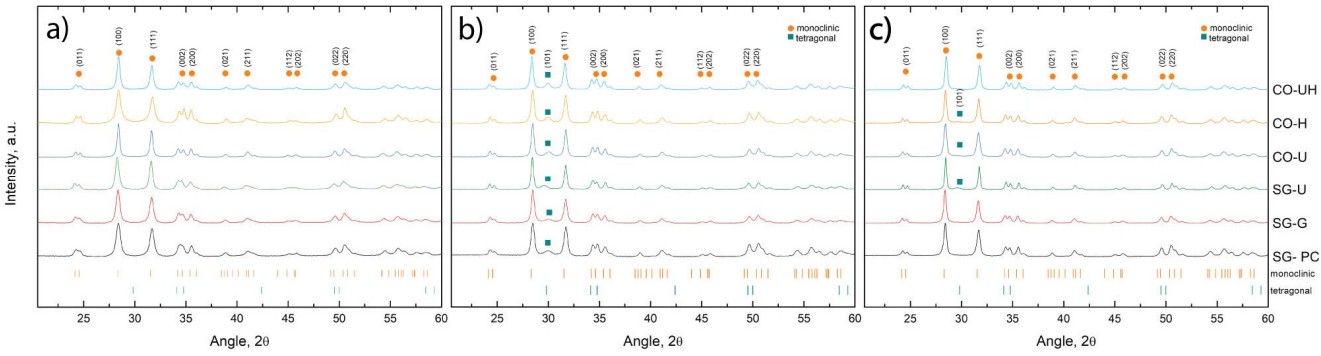

**Figure 1.** X-ray diffraction data for (**a**) undoped HfO$_2$, (**b**) HfO$_2$:Eu, and (**c**) HfO$_2$:Eu for Nb samples.

**Table 1.** Crystallite sizes, tetragonal phase percentage, and Eu and Nb at. % in the doped samples.

| | Crystallite Size, nm | | | % of T Phase | | | at. % of Eu | at. % of Eu | at. % of Nb |
|---|---|---|---|---|---|---|---|---|---|
| Sample | HfO$_2$ | HfO$_2$:Eu | HfO$_2$:Eu, Nb | HfO$_2$ | HfO$_2$:Eu | HfO$_2$:Eu, Nb | HfO$_2$:Eu | HfO$_2$:Eu, Nb | HfO$_2$:Eu, Nb |
| SG-PC | 18.6 | 22.6 | 28.0 | 0.0 | 8.1 | 0.7 | 4.7 | 4.5 | 6.5 |
| SG-G | 22.8 | 26.2 | 30.0 | 0.5 | 6.0 | 1.1 | 4.7 | 4.6 | 2.1 |
| SG-U | 24.1 | 32.4 | 42.5 | 0.3 | 6.5 | 4.4 | 7.0 | 3.7 | 5.5 |
| CO-U | 37.3 | 28.0 | 31.2 | 0.5 | 7.5 | 1.1 | 4.4 | 4.5 | 6.5 |
| CO-H | 17.3 | 23.9 | 26.2 | 2.1 | 8.7 | 1.2 | 4.3 | 4.8 | 6.5 |
| CO-UH | 25.4 | 27.0 | 32.4 | 0.7 | 9.1 | 2.2 | 4.3 | 4.4 | 6.6 |

Crystallite size was determined for all samples using the Scherrer equation, as seen in Table 1. The addition of dopant ions leads to an increase in crystallite size for all methods. While, for the SG group, the crystallite size increased on average by 5.85 nm with each addition of dopants, the CO group showed an average increase of 3.2 nm, indicating that during the sol–gel synthesis, the final structure is either less relaxed than in the

combustion synthesis group or that the addition of dopants during sol–gel synthesis leads to particle coalescence.

EDX data suggest that in the $HfO_2$:Eu samples, the atomic percentage of the Eu ion presence was on average larger for the sol–gel methods (5.5 at. %) than in combustion methods (4.3 at. %). However, in $HfO_2$:Eu of Nb samples, Nb ion at. % in the samples was on average higher for the combustion methods. Overall, the average Eu at. % in $HfO_2$:Eu was 4.9%, in $HfO_2$:Eu, Nb 4.4% and the Nb content was 5.6%.

### 3.2. FTIR

FTIR measurements were performed to determine the organic component presence in the materials. The bonds in the respective regions are shown in Figure 2. We can observe peaks at 777, 541 cm$^{-1}$ due to the formation of Hf-O bonds in which the range of IR (800–400 cm$^{-1}$) are photon modes of crystalline hafnium [27]. Figure 2 reveals that many contamination bonds are visible in the spectra—they are denoted in the graph. Overall, it can be seen that the combustion method leaves fewer organics in the material as compared to the sol–gel method. Adding urea ($CO(NH_2)_2$) during the synthesis leaves residual organic components in the samples, as shown for SG-U and CO-U, where two distinct peaks are seen at around 2000–2200 cm$^{-1}$. These peaks are associated with C≡N and C≡C [28–30]. SG-U also has an intensive peak at 2300 cm$^{-1}$, which is indicated as C≡C [29,31], specifying a significant presence of carbon in the SG-U sample. Doping with the charge compensating Nb$^{5+}$ ions reduces the amount of oxygen vacancies and the 1200–1800 cm$^{-1}$ (C-N, C-C, C-O, C=N, C=C, C=O) and 2800–3800 cm$^{-1}$ (O-H, C-H, N-H) peaks show lower intensity as compared to undoped or Eu-doped $HfO_2$ [28–30]. This suggests that oxygen vacancies have a significant impact on the type and amount of organic bonds present; however, it is also strongly affected by the kinds of organic matter introduced in powder synthesis. The SG-PC sample did not contain nitrogen during the synthesis and therefore it can be deduced that the peaks at 3000 cm$^{-1}$ are more related to C-H rather than C-N.

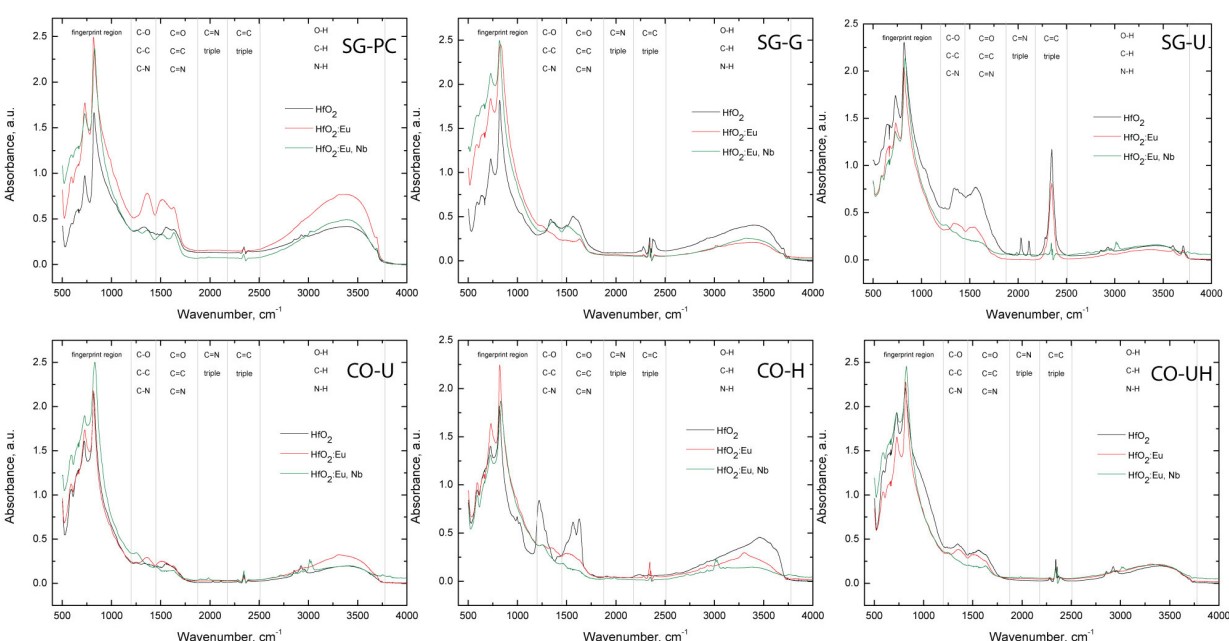

**Figure 2.** FTIR spectra of SG and CO samples.

### 3.3. SEM

In the SEM micrographs (Figure 3), we observe distinct morphological disparities between the sol–gel and combustion-synthesized samples. The sol–gel sample group is composed of irregular nanoparticles with rough grain boundaries, revealing nanostructured

grain formation in the hafnia, characterized by irregular shapes and sizes, whereas the combustion samples present a more homogeneous surface texture and faceted, more regular grains indicating a possibly more controlled growth mechanism. In the sol–gel samples, the nanoparticles appear more as individual particles, while the particles in the combustion samples appear to have coalesced into each other and share common grain boundaries. This morphological distinction may account for the observed difference in photoluminescence properties. Specifically, the reduced integrated PL intensity for the undoped combustion samples could be attributed to the relatively smaller amount of intrinsic defects on their comparatively smoother surface and intergranular boundaries.

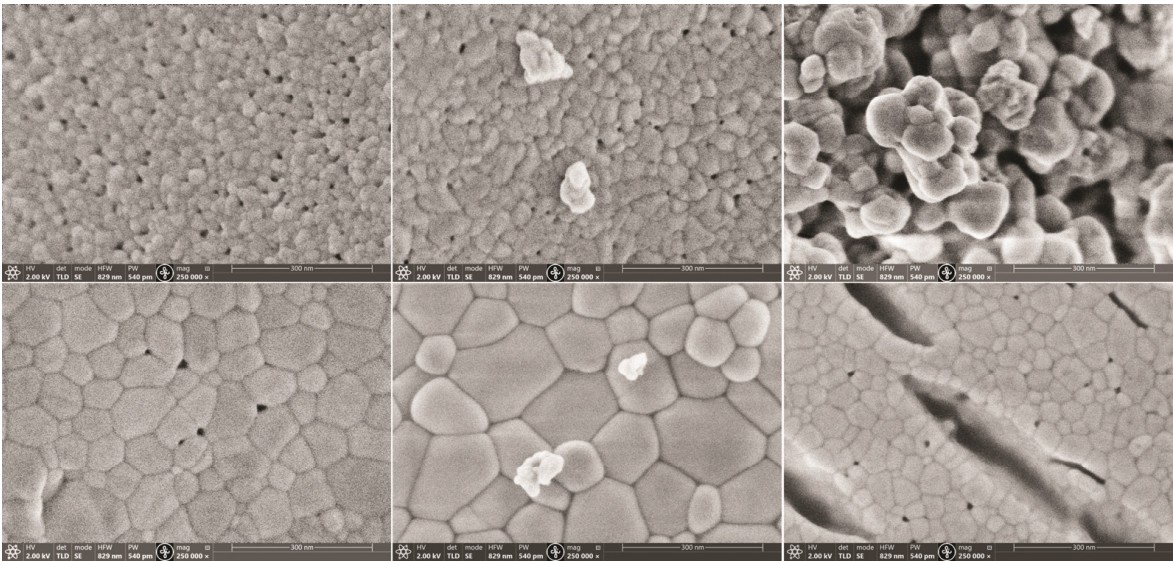

**Figure 3.** SEM micrographs for the undoped SG and CO samples.

### 3.4. Photoluminescence

PL studies were used to correlate the structural properties of the powders to the spectra. The normalized photoluminescence spectra (Figure 4) show a standard intrinsic defect luminescence band for the undoped samples. The spectra showed similar characteristics, confirming the overall quality of the samples. Peak maxima were located between 2.38 eV (CO-H) and 2.41 eV (SG-G). Due to the wide band gap, the electrons in the valence band are not directly excited to the conduction band. The electrons can only be excited via the charge transfer band, which is located below the conduction band. The relaxation of excited electrons to the valence band happens through different defect levels, leading to a broad emission spectrum. The defects that capture the electrons are mainly threefold- and fourfold-coordinated single- and double-charged oxygen vacancies. Electrons are captured in the oxygen vacancy defects and are further distributed over the nearest Hf atom [32], leading to a broad emission spectrum. The primitive unit cell of m-$HfO_2$ consists of 12 atoms including 4 Hf atoms and 8 O atoms. The Hf atoms exhibit sevenfold coordination, having two kinds of oxygen atoms with different coordination: $O_3$ having three Hf neighbors and $O_4$ having four Hf neighbors [24,33,34].

Thus two types of O vacancies exist in m-$HfO_2$: threefold- and fourfold-coordination vacancies, and the spectra of undoped samples reveal that SG-G and CO-UH samples have a slightly increased number of fourfold-coordinated vacancies if compared to other atoms.

The characteristic Eu luminescence in the doped samples for both methods of synthesis can be seen in Figure 5. The obtained spectra are highly similar for all the samples, confirming successful dopant incorporation into the host matrix. Studies show that the intensity of the PL often correlates to the concentration of oxygen vacancies in the sample powders [24,35]. To evaluate and compare the PL intensities of the samples, the spectra were integrated and the intensity values are shown in Figure 6. Overall, the highest integrated PL

intensity values were seen for samples doped additionally with Nb, confirming successful charge compensation by the $Nb^{5+}$ ions and resulting in the reduction of the number of defects present in the material. Due to the high PL intensity values for Nb-doped samples, the displayed data points need to be multiplied by 6 to achieve the real values.

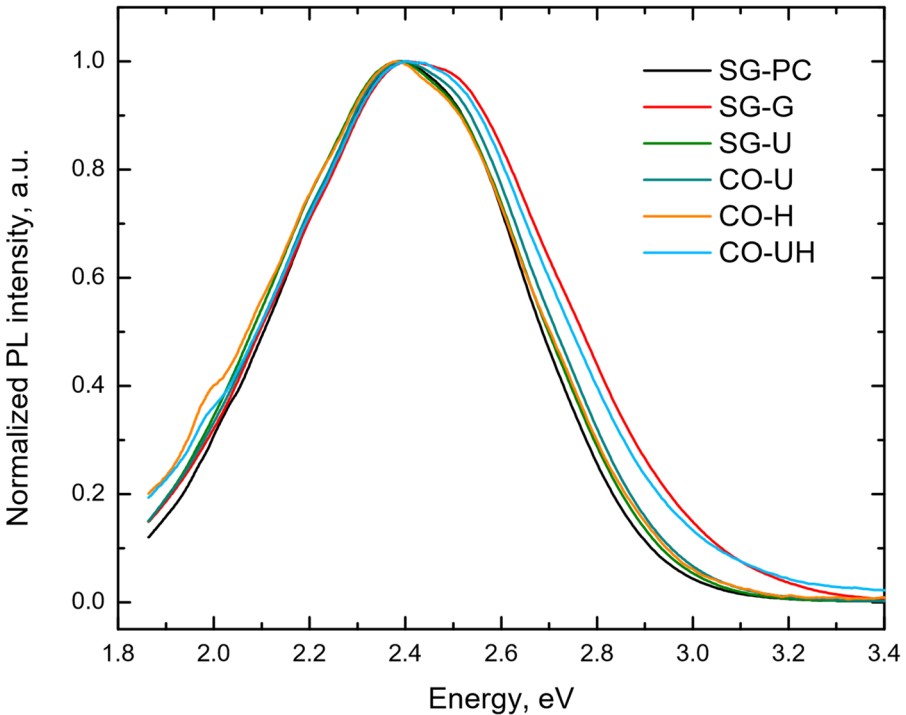

**Figure 4.** Photoluminescence spectra of the undoped $HfO_2$ samples, excited at 266 nm.

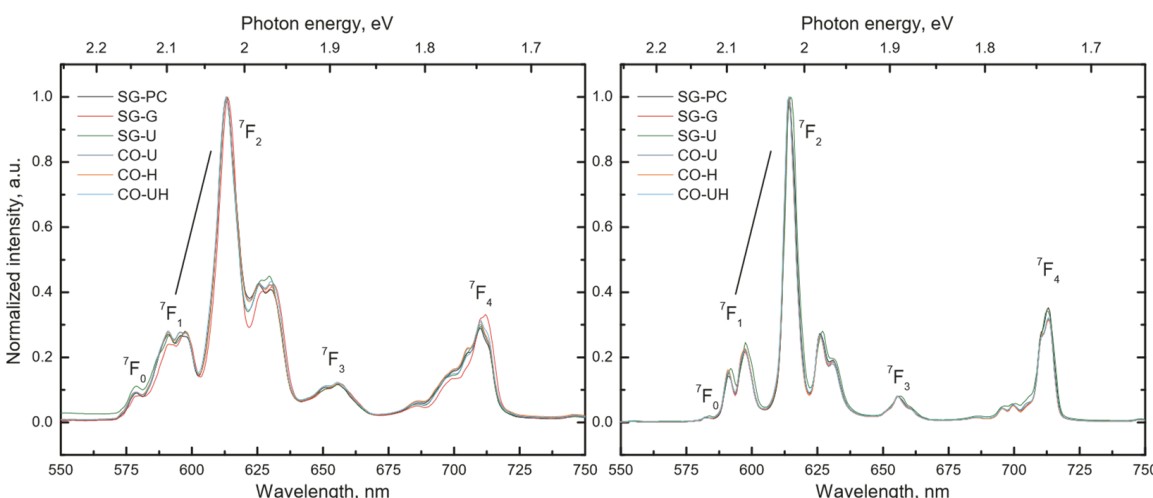

**Figure 5.** Photoluminescence spectra of the $Eu^{3+}$, $Eu^{3+}$, and $Nb^{5+}$-doped $HfO_2$ samples excited at 266 nm.

For undoped samples, the PL intensity for the SG samples was the highest, combined with the highest luminescence intensity when doped with $Eu^{3+}$. Conversely, CO samples showed low luminescence intensity for the undoped $HfO_2$ and lower luminescence intensity for $Eu^{3+}$-doped samples.

Additionally, $HfO_2$:Eu, Nb SG samples showed a significantly lower luminescence intensity than $HfO_2$:Eu, Nb CO samples. It appears that a lower PL intensity in the undoped

samples leads to a lower PL intensity in the Eu$^{3+}$-doped samples; however, the Eu$^{3+}$ and Nb$^{5+}$-doped samples show the opposite behavior.

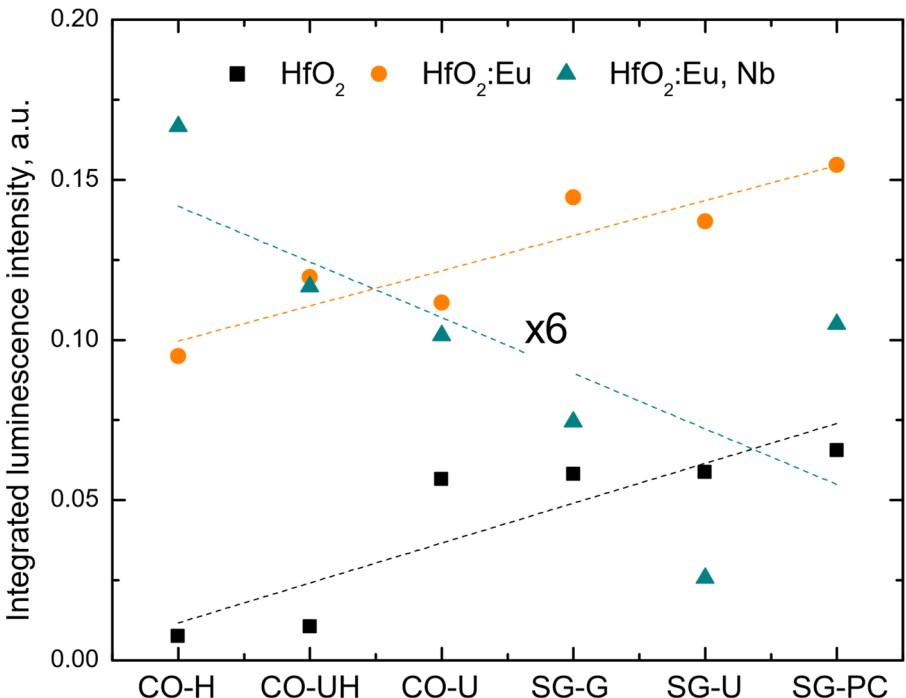

**Figure 6.** Integrated photoluminescence intensities of the undoped, Eu$^{3+}$, Eu$^{3+}$, and Nb$^{5+}$-doped HfO$_2$ samples excited at 266 nm.

*3.5. XPS*

Peak fitting of the Hf4f spectra was carried out using the Shirley background subtraction and Gaussian/Lorentzian functions. The results are represented by two pairs of Gaussian lines (doublet) corresponding to 4f5/2 and 4f7/2 sublevels of the 4f level due to spin–orbit splitting one to stoichiometric HfO$_2$ and the other one to oxygen-deficient HfO$_{2-x}$ (Figure 7, doublet denoted by 1—HfO$_2$ and 2—HfO$_{2-x}$) [36–40]. The literature states that it is not uncommon for hafnium suboxide to be formed in addition to the hafnium dioxide [22,41,42].

We can observe that, for some samples, the doublet peaks of the suboxidized hafnium are stronger than those of the fully oxidized Hf$^{4+}$, and for some—weaker. The width of the peaks is broader for the suboxide peaks, indicating the presence of a higher degree of disorder in the oxygen-deficient hafnia species (see Appendix A Table A1) [43]. There is also an observable shift in binding energy (BE) when doping the samples. Several factors can lead to binding energy shifts, such as the charge transfer effect, environmental charge density, the presence of an electric field, and hybridization [44]. Additionally, the response of the material, when electrons are localized on a vacancy, can influence the observed binding energy as well. Typically, more positively charged environments lead to higher BEs, while more negatively charged environments result in lower BEs [45]. Oxygen vacancies in metal oxides introduce a positively charged environment; therefore, the samples with more oxygen vacancies would exhibit a shift in BE position due to the change in the electronic environment around the Hf sites and charge compensation mechanisms. It is clearly visible that doping with trivalent Eu ions leads to an increase in the vacancies; however, doping with additional Nb$^{5+}$ leads to mitigation of this effect. Also, it can be noted that SG-U and CO-H as well as CO-UH samples exhibit a large amount of vacancies.

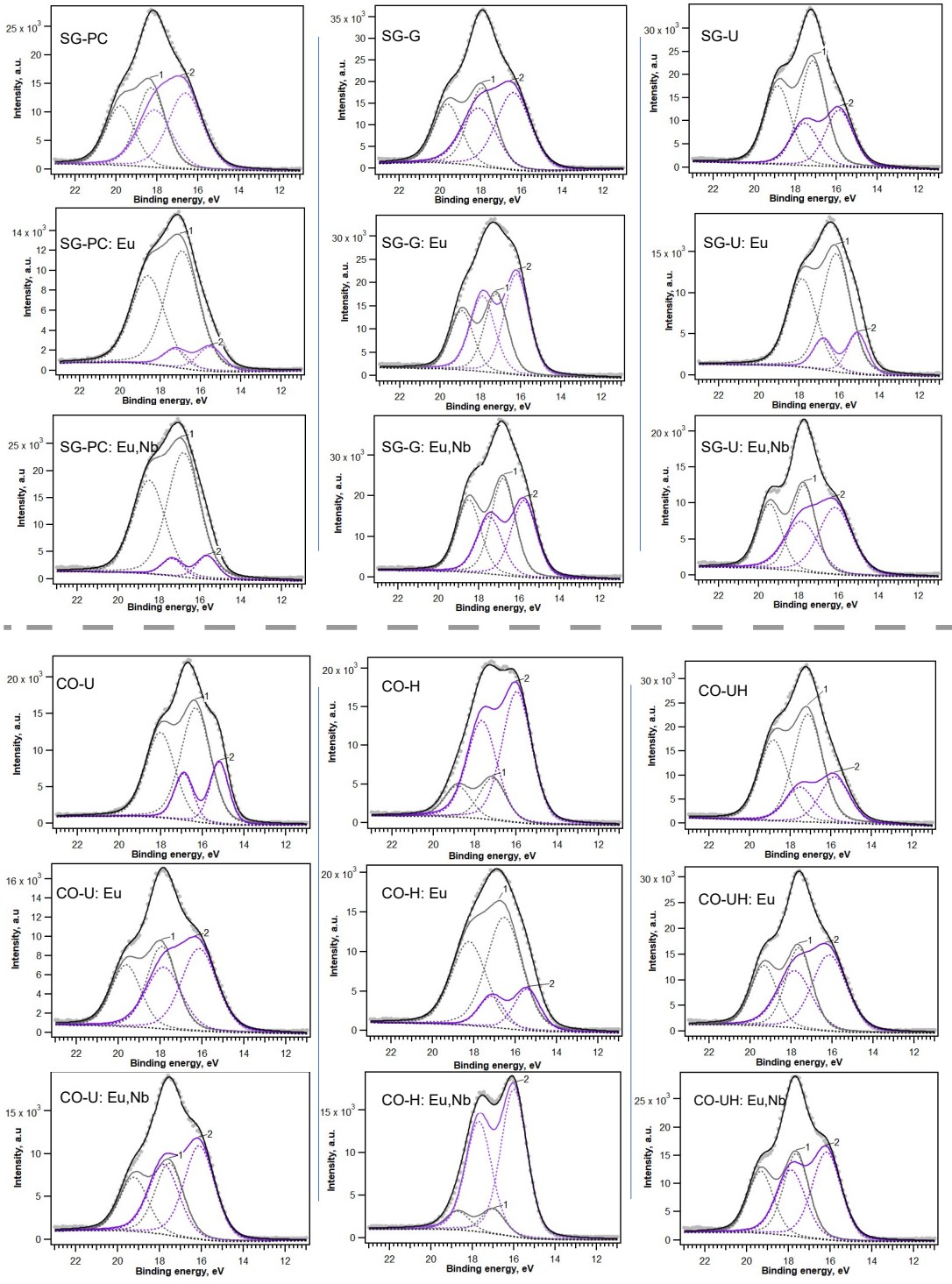

**Figure 7.** XPS spectra of the SG and CO sample groups. Line 1—HfO$_2$ and line 2—HfO$_{2-x}$.

As seen in Figure 8, while the undoped SG and CO samples typically exhibit similar levels of suboxides, the introduction of Eu$^{3+}$ as a dopant results in a substantial increase in suboxides in SG samples. Conversely, CO samples experience only a slight increase in suboxide content on average. Further doping with Nb leads to a significant reduction in the number of suboxides within the combustion group, affirming a decrease in oxygen vacancies due to the charge-compensating influence of Nb$^{5+}$. However, the reduction in the suboxide species in the SG samples is less pronounced, consistent with the earlier-observed growth in crystallite sizes and a non-relaxed structure.

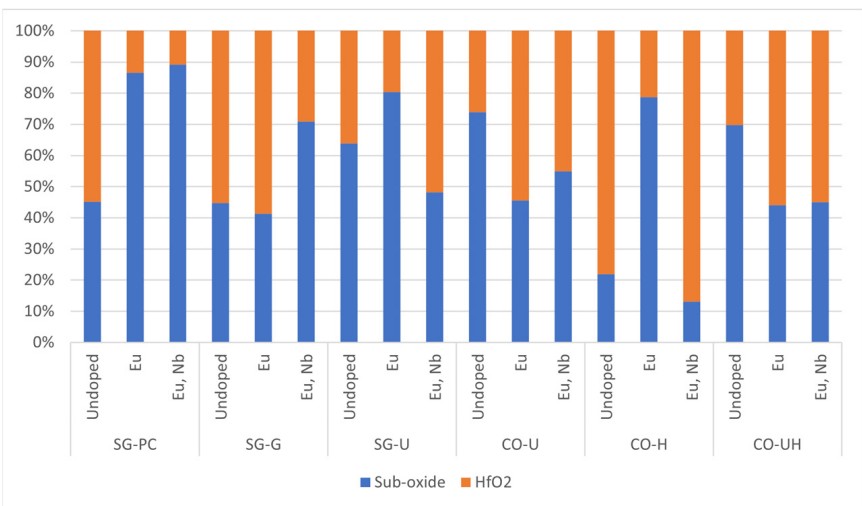

**Figure 8.** Percentage of suboxides in the SG and CO sample groups.

## 4. Conclusions

This research has presented a comprehensive analysis of synthesizing $HfO_2$ using both sol–gel and combustion methods and further detailed the incorporation of europium ($Eu^{3+}$) and niobium ($Nb^{5+}$) as dopants to monitor defects in the resulting materials. The sol–gel-derived $HfO_2$ exhibited nanoparticles with irregular shapes and distinct intergranular boundaries. In contrast, combustion synthesis resulted in a homogeneous morphology with an evident trend towards the coalescence of particles, suggesting a controlled growth mechanism.

It was confirmed that doping with $Eu^{3+}$ led to the creation of oxygen vacancies which, in turn, stabilized the tetragonal phase; however, all samples remained largely monoclinic. The introduction of $Nb^{5+}$ as an additional dopant acted as a charge-compensating agent, reducing oxygen vacancies significantly. For the sol–gel samples, the addition of dopants led to a larger increase in average crystallite sizes than for combustion samples, indicating that the final SG structure is either less relaxed or that the addition of dopants during sol–gel synthesis leads to particle coalescence.

Moreover, the atomic percentage of the Eu ions was found to be higher in sol–gel methods as compared to combustion techniques. The combustion synthesis method seems to leave fewer organic residues as compared to the sol–gel technique.

Notably, the presence of urea during synthesis was identified as a contributing factor to the residual organic components in $HfO_2$. Photoluminescence analysis confirms that doping with $Nb^{5+}$ significantly reduced the amount of oxygen vacancies, thereby enhancing the visible $Eu^{3+}$ emission. The XPS peak fitting of Hf4f spectra revealed distinct features for stoichiometric $HfO_2$ and its oxygen-deficient counterpart, $HfO_{2-x}$. Observable differences in peak intensity, width, and shifts in binding energy highlight the presence of oxygen vacancies and their influence on the electronic environment surrounding the Hf sites.

**Author Contributions:** Conceptualization, K.L., V.V., I.B. and A.Z.; data curation, K.L. and V.V.; funding acquisition, K.S.; investigation, K.L., V.V., I.B., E.E., M.L. and G.T.; methodology, K.L., I.B., K.S. and A.Z.; supervision and project administration, K.S. and A.Z.; writing—original draft, K.L. and I.B.; writing—review and editing, K.L., I.B. and V.V. All authors have read and agreed to the published version of the manuscript.

**Funding:** The financial support of the European Regional Development Fund (ERDF) Project No. 1.1.1.1/21/A/055 is greatly acknowledged. Institute of Solid State Physics, University of Latvia as the Center of Excellence has received funding from the European Union's Horizon 2020 Framework Programme H2020-WIDESPREAD-01-2016-2017-TeamingPhase2 under grant agreement no. 739508, project CAMART2.

**Institutional Review Board Statement:** Not applicable.

**Informed Consent Statement:** Not applicable.

**Data Availability Statement:** The data presented in this study are available on request from the corresponding author.

**Conflicts of Interest:** The authors declare no conflict of interest.

## Abbreviations

The following abbreviations are used in this manuscript:

| | |
|---|---|
| XRD | X-ray diffraction |
| EDX | Energy Dispersive X-ray Spectroscopy |
| FTIR | Fourier Transform Infrared Spectroscopy |
| PL | Photoluminescence |
| XPS | X-ray Photoelectron Spectroscopy |
| BE | Binding energy |
| SG | Sol–gel synthesis |
| SG-PC | Sol–gel synthesis—polymerized complex |
| SG-G | Sol–gel synthesis—glycine |
| SG-U | Sol–gel synthesis—urea |
| CO | Combustion synthesis |
| CO-U | Combustion synthesis—urea |
| CO-H | Combustion synthesis—hexamine |
| CO-UH | Combustion synthesis—urea and hexamine |

## Appendix A

**Table A1.** XPS peak data—peak position, eV; area, a.u.; width coefficient.

| | | | |
|---|---|---|---|
| SG-PC | Peak position, eV<br>area, a.u.<br>width coefficient | 18.18, 19.88<br>26,687, 20,015<br>1.66 | 16.67, 18.37<br>32,354, 24,266<br>2.00 |
| SG-PC:Eu | Peak position, eV<br>area, a.u.<br>width coefficient | 16.93, 18.63<br>27,435, 576<br>1.98 | 15.53, 17.23<br>4269, 3201<br>1.46 |
| SG-PC:Eu,Nb | Peak position, eV<br>area, a.u.<br>width coefficient | 16.80, 18.50<br>51,130, 38,347<br>1.88 | 15.66, 17.36<br>6154, 4616<br>1.20 |
| SG-G | Peak position, eV<br>area, a.u.<br>width coefficient | 17.94, 19.64<br>34,183, 25,637<br>1.59 | 16.40, 18.10<br>42,236, 31,677<br>2.00 |
| SG-G:Eu | Peak position, eV<br>area, a.u.<br>width coefficient | 17.24, 18.94<br>26,597, 19,947<br>1.35 | 16.22, 17.92<br>37,766, 28,324<br>1.46 |
| SG-G:Eu,Nb | Peak position, eV<br>area, a.u.<br>width coefficient | 16.67, 18.37<br>50,984, 38,238<br>1.54 | 15.59, 17.29<br>20,897, 15,673<br>1.27 |
| SG-U | Peak position, eV<br>area, a.u.<br>width coefficient | 17.16, 18.86<br>40,717, 30,538<br>1.52 | 15.87, 17.57<br>23,084, 17,313<br>1.60 |
| SG-U:Eu | Peak position, eV<br>area, a.u.<br>width coefficient | 16.15, 17.85<br>28,348, 21,261<br>1.69 | 15.10, 16.80<br>6904, 5178<br>1.18 |
| SG-U:Eu,Nb | Peak position, eV<br>area, a.u.<br>width coefficient | 17.75, 19.45<br>19,962, 14,971<br>1.41 | 16.17, 17.87<br>21,515, 16,136<br>1.97 |

**Table A1.** *Cont.*

| CO-U | Peak position, eV<br>area, a.u.<br>width coefficient | 16.33, 18.03<br>30,368, 22,776<br>1.66 | 15.20, 16.90<br>10,752, 8064<br>1.07 |
|---|---|---|---|
| CO-U:Eu | Peak position, eV<br>area, a.u.<br>width coefficient | 17.94, 19.64<br>16,899, 12,674<br>1.67 | 16.15, 17.85<br>20,209, 15,156<br>2.00 |
| CO-U:Eu,Nb | Peak position, eV<br>area, a.u.<br>width coefficient | 17.54, 19.24<br>26,959, 11,886<br>1.59 | 16.10, 17.80<br>22,091, 16,568<br>1.73 |
| CO-H | Peak position, eV<br>area, a.u.<br>width coefficient | 17.14, 18.84<br>9476, 7107<br>1.48 | 15.97, 17.67<br>33,798, 25,348<br>1.69 |
| CO-H:Eu | Peak position, eV<br>area, a.u.<br>width coefficient | 16.54, 18.24<br>33,211, 24,908<br>2.00 | 15.45, 17.15<br>9002, 6751<br>1.46 |
| CO-H:Eu,Nb | Peak position, eV<br>area, a.u.<br>width coefficient | 17.02, 18.72<br>4578, 3433<br>1.34 | 16.00, 17.70<br>30,324, 22,743<br>1.49 |
| CO-UH | Peak position, eV<br>area, a.u.<br>width coefficient | 17.14, 18.84<br>44,377, 33,283<br>1.68 | 15.83, 17.53<br>19,270, 14,453<br>1.71 |
| CO-UH:Eu | Peak position, eV<br>area, a.u.<br>width coefficient | 17.64, 19.34<br>27,483, 20,612<br>1.49 | 16.15, 17.85<br>34,975, 26,231<br>2.00 |
| CO-UH:Eu,Nb | Peak position, eV<br>area, a.u.<br>width coefficient | 17.64, 19.34<br>24,742, 18,556<br>1.43 | 16.18, 17.88<br>30,263, 22,697<br>1.65 |

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
