# Peer review of "Impact of Europium and Niobium Doping on Hafnium Oxide (HfO2): Comparative Analysis of Sol–Gel and Combustion Synthesis Methods"

_ceramics, doi:10.3390/ceramics7010002_

Round 1

Reviewer 1 Report

Comments and Suggestions for Authors

This paper study the difference in morphology and photoluminescence properties of HfO2 material derived from two different methods. Some new and interesting information can be found. This paper can be accepted after addressing the following issues.

1. In the section "2. Materials and Methods": Although the relevant preparation methods are referenced, the used raw  materials  and the preparation processes should be summarized for the convenience of the reader. The solution concentration and the calcination temperature for powder synthesis should be stated.

2.  The full name of HMTA should be given.

3.  The details on FTIR measurements was not mentioned in Section "2.3 Characterization".

4. The authors claim that doping induced oxygen vacancies lead to the form of tetragonal phase HfO2. More evidence is needed to support this conclusion. In fact, the crystal structure of metal ion doped oxides is often affected by lattice distortion caused by the difference in the radius of doped ions. To clarify this problem, it is necessary to consider the effect of ion doping on the lattice constants of HfO2.

5. In Section "3.2 FTIR": "This suggests that oxygen vacancies have a significant impact on the type and amount of organic bonds present." The presence of residual organic matter is more affected by the kinds of organic matter introduced in powder synthesis and the calcination temperature.

6. In order to accurately analyze the change of oxygen vacancy concentration caused by doping, accurate thermogravimetric analysis and iodometric titration tests are required.

7. The samples corresponding to in Fig. 3 are synthesized powders, or  pressed green or sintered ceramic pellets? The structure in the figures looks like that of sintered ceramics. In order to better compare the effect of the synthesis methods and related auxiliary agents on the morphology of the synthesized powders, the dispersed powders should be used for high magnification SEM observation.

8. The title is "Tailoring HfO2 Properties: A Comparative Study of Combustion

and Sol-Gel Synthesis". However,  on the whole, this paper failed to clearly clarify the direct effects of synthesis methods and related auxiliary agents (glycine, urea, hexamine, and etc. ),  on the performance of HfO2 and the corresponding mechanism, but mainly discussed the influence of ion doping.

Comments on the Quality of English Language

Minor editing of English language is required.

Author Response

Dear Reviewer, thank you for Your time and valuable suggestions! We have carefully examined the comments and provide our answers and corrections:

Reviewer #1

  1. In the section "2. Materials and Methods": Although the relevant preparation methods are referenced, the used raw  materials  and the preparation processes should be summarized for the convenience of the reader. The solution concentration and the calcination temperature for powder synthesis should be stated.

We have updated the manuscript with necessary information.

  1. The full name of HMTA should be given.

We have updated the manuscript with necessary information.

  1. The details on FTIR measurements was not mentioned in Section "2.3 Characterization".

We have updated the manuscript with necessary information.

  1. The authors claim that doping induced oxygen vacancies lead to the form of tetragonal phase HfO2. More evidence is needed to support this conclusion. In fact, the crystal structure of metal ion doped oxides is often affected by lattice distortion caused by the difference in the radius of doped ions. To clarify this problem, it is necessary to consider the effect of ion doping on the lattice constants of HfO2.

Here we use XRD measurements to confirm the presence of tetragonal phase HfO2. Fischer et.al states that there is no clear correlation visible between the dopant  ionic radius and the stabilization efficiency of the tetragonal phase.  [The Effect of Dopants on the Dielectric Constant of HfO2 and ZrO2 from First Principles, January 2008 Applied Physics Letters 92(1):012908-012908-3 DOI: 10.1063/1.2828696]. However, in another publication  [Fischer, D., & Kersch, A. (2008). Stabilization of the high-k tetragonal phase in HfO2: The influence of dopants and temperature from ab initio simulations. Journal of Applied Physics, 104(8), 084104. doi:10.1063/1.2999352] the author  states that larger dopant concentrations are needed for the tetragonal phase stabilisation. Therefore we concluded that the effect might come from the oxygen vacancy distorted lattice. Other authors have arrived to similar conclusions, i.e. [Origin of Ferroelectric Phase in Undoped HfO2 Films Deposited by Sputtering, "Advanced electronic materials". 2019, 6(11), Art.-Nr. 1900042. ISSN 2199-160X https://doi.org/10.1002/admi.201900042]

  1. In Section "3.2 FTIR": "This suggests that oxygen vacancies have a significant impact on the type and amount of organic bonds present." The presence of residual organic matter is more affected by the kinds of organic matter introduced in powder synthesis and the calcination temperature.

We have added this remark to the manuscript.

  1. In order to accurately analyze the change of oxygen vacancy concentration caused by doping, accurate thermogravimetric analysis and iodometric titration tests are required.

We agree that these methods would provide valuable insights and enhance the robustness of our findings. However, due to current constraints in time and resources, it is challenging for us to incorporate these tests in the present study. We recognize the importance and potential impact of these analyses. Therefore, we are considering them for a future study where we can dedicate the appropriate resources and time to conduct these tests thoroughly.

  1. The samples corresponding to in Fig. 3 are synthesized powders, or  pressed green or sintered ceramic pellets? The structure in the figures looks like that of sintered ceramics. In order to better compare the effect of the synthesis methods and related auxiliary agents on the morphology of the synthesized powders, the dispersed powders should be used for high magnification SEM observation. 

These are sintered ceramic pellets. We chose to test the pellets, as all other analyses are for the ceramics rather than the powder materials.

  1. The title is "Tailoring HfO2 Properties: A Comparative Study of Combustion and Sol-Gel Synthesis". However,  on the whole, this paper failed to clearly clarify the direct effects of synthesis methods and related auxiliary agents (glycine, urea, hexamine, and etc. ),  on the performance of HfO2 and the corresponding mechanism, but mainly discussed the influence of ion doping.

Following your suggestion, we changed the name of the manuscript to “Impact of Europium and Niobium Doping on Hafnium Oxide (HfO2): Comparative Analysis of Sol-Gel and Combustion Synthesis Methods”

Reviewer 2 Report

Comments and Suggestions for Authors

Dear authors,

This manuscript is nicely written and may interest the Journal Ceramics readership. You correctly described and presented the results of your research. However, it can only be accepted for publication after a minor revision. Here are my comments.

Comment 1: Line 19 …,, It has a wide band gap (> 5 eV), high refractive index...”

In this sentence, you are missing two references for band gap and refractive index, so please provide them.

Comment 2: Line 24 …,, HfO2 exists in different polymorph phases - monoclinic, cubic, tetragonal, and orthorhombic....”

Please change the order according to the degree of arrangement of the structures - cubic, tetragonal, orthorhombic, and monoclinic and put the number of the space group in parentheses behind each one, for example (SG 137). It would also be nice, just because of the comparison of the obtained powders in relation to the synthesis, if you can find at what temperature which modification is formed. For example, the transition from the tetragonal to the cubic phase occurs at 1480 0 C.

Comment 3: lines 34-35…,, also doped with Eu3+ ions known to create oxygen vacancies....”

Please add a reference.

Comment 4: lines 35-36…,, doped with Nb5+ ions which act as a charge compensating element that in turn reduces the amount of oxygen vacancies…”

Please add a reference.

Comment 5: lines 37-38…,, Eu3+ ions are also known as luminescent probes...”

Please add a reference.

Comment 6: line 38 Please, at the end of this section, add another sentence about why you specifically researched satisfactory syntheses to obtain that particular material with specific properties that you would use for what? What is the main goal of this manuscript?

Comment 7: Dear colleagues, I have a well-intentioned proposal for you. After trying many syntheses and using different fuels if you get the chance and have the time apply propellant chemistry. It is a different way of calculating the chemicals that you will use in the synthesis. The point is to get the particle size below 20nm. Only then will they show those other properties that we need for the application of new materials. Also, please establish the exact sintering temperature. Use dilatometric measurements. If you have time read this:

K.C. Patil, M.S. Hegde, T. Rattan, S.T. Aruna, Chemistry Of Nanocrystalline Oxide Materials: Combustion Synthesis: Properties and Applications, World Scientific Publishing Co., Pte. Ltd., Singapore, 2008.

https://doi.org/10.1016/j.ceramint.2010.12.015

Comment 8: Line 62 Please state the recording conditions, the base you used to identify the phases and their numbers, and the space group number e.g. ICSD #173966, SG 137.

Comment 9: Please change Figure 1: put Intensity (a.u.) on the ordinate and on the abscissa 2Ɵ / 0. To identify the phase, you need at least three peaks corresponding to it. Label the 3 reflections for the tetragonal phase with the corresponding Miller indices. If the peaks of the monoclinic and tetragonal phases match, choose the one with higher intensity. Remove the orange ball at the 37 for 2Ɵ / 0. Who does reflection 112 belong to? Mark the peaks of 54-59 for 2Ɵ / 0.

Comment 10: Line 100 …,, FTIR measurements were performed to determine the organic component presence in the materials. ...”

Why, when FTIR measurements used to determine the presence of both organic and inorganic components in materials? In your case, we are interested in Eu and Nb dopants and of course Hf. Therefore, please pay attention to the regions of 2800-3100 and 3600-3800 cm-1 and give us an explanation for Hf, Eu, and Nb in this section.

Comment 11: Line 123 Please add a sentence describing the grains, shape, form...

Author Response

Dear Reviewer, thank you for Your time and valuable suggestions! We have carefully examined the comments and provide our answers and corrections:

Reviewer #2

Comment 1: Line 19 …,, It has a wide band gap (> 5 eV), high refractive index...”

In this sentence, you are missing two references for band gap and refractive index, so please provide them.

We have updated the document with the missing references.

Comment 2: Line 24 …,, HfO2 exists in different polymorph phases - monoclinic, cubic, tetragonal, and orthorhombic....”

Please change the order according to the degree of arrangement of the structures - cubic, tetragonal, orthorhombic, and monoclinic and put the number of the space group in parentheses behind each one, for example (SG 137). It would also be nice, just because of the comparison of the obtained powders in relation to the synthesis, if you can find at what temperature which modification is formed. For example, the transition from the tetragonal to the cubic phase occurs at 1480 0 C.

We have updated the manuscript with the necessary information.

Comment 3: lines 34-35…,, also doped with Eu3+ ions known to create oxygen vacancies....”

Please add a reference.

We have updated the document with the missing references.

Comment 4: lines 35-36…,, doped with Nb5+ ions which act as a charge compensating element that in turn reduces the amount of oxygen vacancies…”

Please add a reference.

We have updated the document with the missing references.

Comment 5: lines 37-38…,, Eu3+ ions are also known as luminescent probes...”

Please add a reference.

We have updated the document with the missing references.

Comment 6: line 38 Please, at the end of this section, add another sentence about why you specifically researched satisfactory syntheses to obtain that particular material with specific properties that you would use for what? What is the main goal of this manuscript?

We have added this sentence: 

Both methods are popular for synthesizing advanced ceramics and have unique char-acteristics, and with this publication we want to emphasize that the method of synthe-sis plays a crucial role for amount of oxygen vacancies, structure of obtained ceramics and, in turn, the usability of the ceramics.

Comment 7: Dear colleagues, I have a well-intentioned proposal for you. After trying many syntheses and using different fuels if you get the chance and have the time apply propellant chemistry. It is a different way of calculating the chemicals that you will use in the synthesis. The point is to get the particle size below 20nm. Only then will they show those other properties that we need for the application of new materials. Also, please establish the exact sintering temperature. Use dilatometric measurements. If you have time read this:

K.C. Patil, M.S. Hegde, T. Rattan, S.T. Aruna, Chemistry Of Nanocrystalline Oxide Materials: Combustion Synthesis: Properties and Applications, World Scientific Publishing Co., Pte. Ltd., Singapore, 2008.

https://doi.org/10.1016/j.ceramint.2010.12.015

 Thank you very much for your comment! We appreciate it and will take note for our future research.

Comment 8: Line 62 Please state the recording conditions, the base you used to identify the phases and their numbers, and the space group number e.g. ICSD #173966, SG 137.

We updated the information. The space groups were mentioned before, so here we decided to not repeat.

Comment 9: Please change Figure 1: put Intensity (a.u.) on the ordinate and on the abscissa 2Ɵ / 0. To identify the phase, you need at least three peaks corresponding to it. Label the 3 reflections for the tetragonal phase with the corresponding Miller indices. If the peaks of the monoclinic and tetragonal phases match, choose the one with higher intensity. Remove the orange ball at the 37 for 2Ɵ / 0. Who does reflection 112 belong to? Mark the peaks of 54-59 for 2Ɵ / 0.

We have changed Figure 1 according to Your suggestion.

Comment 10: Line 100 …,, FTIR measurements were performed to determine the organic component presence in the materials. ...”

Why, when FTIR measurements used to determine the presence of both organic and inorganic components in materials? In your case, we are interested in Eu and Nb dopants and of course Hf. Therefore, please pay attention to the regions of 2800-3100 and 3600-3800 cm-1 and give us an explanation for Hf, Eu, and Nb in this section.

The fingerprint region of hafnia is located under 1000 cm-1. We have added information about that. The regions that you mention do not contain useful information about hafnia vibrations.

Comment 11: Line 123 Please add a sentence describing the grains, shape, form...

Thank you for the suggestion, the paragraph is now corrected to look like this:

The sol-gel sample group is composed of irregular nanoparticles with rough grain boundaries, revealing nanostructured grain formation in the hafnia, characterized by irregular shapes and sizes. whereas the combustion samples present a more homogeneous surface texture and faceted, more regular grains indicating a possibly more controlled growth mechanism. In the sol-gel samples, the nanoparticles appear more as individual particles, while the particles in the combustion samples appear to have coalesced into each other and share common grain boundaries. 

Round 2

Reviewer 1 Report

Comments and Suggestions for Authors

After revision, the manuscript is improved to some degree. There are still two problems:

1. the sintering conditions for the pressed samples should provided in the Experimental section, since the sintered samples were used for some measurements. 

2. It is strange that the structure of the paper was changed in this revised version. Section 3 was modified as "3. Results", and an additional Section "4 Discussion" was added. However, the added Section 4 has no text of substantive discussion. 

Author Response

  1. the sintering conditions for the pressed samples should provided in the Experimental section, since the sintered samples were used for some measurements.

This information has been already been included in the text “samples were pressed into 5 mm diameter green ceramic pellets with a thickness of 1 mm.”. “Green ceramics” here signify that there was no sintering after the samples were pressed into pellets. Powder sintering conditions have been described in the synthesis descriptions.

  1. It is strange that the structure of the paper was changed in this revised version. Section 3 was modified as "3. Results", and an additional Section "4 Discussion" was added. However, the added Section 4 has no text of substantive discussion.

Thank you for the careful reading - that was a mistake, where a placeholder from the template was mistakenly left undeleted. It is deleted now.
